# The Role of Routine Culture in the Treatment of Chronic Suppurative Otitis Media: Implications for the Standard of Care in Rural Areas of South Africa

**DOI:** 10.3390/tropicalmed4010010

**Published:** 2019-01-08

**Authors:** Julia Toman, Anthony Moll, Melynda Barnes, Sheela Shenoi, J. Zachary Porterfield

**Affiliations:** 1Department of Surgery, Section of Otolaryngology, Yale University School of Medicine, New Haven, CT 06510, USA; juliatoman@gmail.com (J.T.); melyndabarnes@gmail.com (M.B.); 2Antiretroviral Programme, Church of Scotland Hospital, Tugela Ferry 3010, KwaZulu-Natal, South Africa; 4tonymoll@gmail.com; 3Department of Internal Medicine, Section of Infectious Diseases, AIDS Program, Yale University School of Medicine, New Haven, CT 06510, USA; sheela.shenoi@yale.edu; 4School of Public Health, Yale University, New Haven, CT 06510, USA; 5Department of Infectious Diseases, University of KwaZulu-Natal, Durban 4000, South Africa; 6Africa Health Research Institute, Durban 4013, South Africa

**Keywords:** Chronic Suppurative Otitis Media, otorrhea, HIV, tuberculosis, antibiotic resistance, MRSA, methicillin resistant *Staphylocococcus aureus*, *Scedosporium*

## Abstract

Chronic Suppurative Otitis Media (CSOM) is a widely prevalent disease, which is a leading cause of acquired deafness worldwide, and is associated with complications with significant mortality and morbidity. It often responds poorly to standard of care therapy and places a disproportionate burden on at-risk populations. The microbiology and antibiotic resistance of CSOM varies based on local factors, including health care access, comorbidities, and antibiotic prescribing practices. We evaluated the role and feasibility of using routine culture for the treatment of CSOM in rural areas as a means of improving treatment of CSOM. More than 400 patients were screened in a rural clinic in South Africa over six weeks, and 14 met study criteria and consented for participation. Gram-negative organisms predominated overall, although *Staphylococcus aureus* was the most commonly isolated single species. A majority of the pathogens were relatively sensitive to commonly prescribed antibiotics, but two cases of methicillin resistant *Staphylococcus aureus* were cultured, and one patient grew a *Scedosporium* species. Treatment on follow-up was able to be directed by culture results, suggesting routine culture at the initial point of contact with the health care system may play a pivotal role in addressing this widely prevalent and devastating disease.

## 1. Introduction

The World Health Organization estimates that between 65 and 330 million people worldwide are affected by Chronic Suppurative Otitis Media (CSOM), with a prevalence ranging from ~0.9% of children in the developed world to ~20% in the developing world and in certain at-risk populations [1,2,3]. Risk factors for development of CSOM include overcrowding, poor hygiene, malnutrition, frequent upper respiratory tract infections, and poor access to healthcare [2].

Otorrhoea can be a result of an either external ear or middle ear infection, and predominantly affects children. Chronic otorrhea is classically associated with *Pseudomonas aeruginosa*, *Staphylococcus aureus*, and *Klebsiella* spp. infection; however, the responsible pathogens may vary depending on local prevalence, comorbidities (including HIV infection), and patterns of antibiotics use [1,2,3,4,5,6].

Serious complications of CSOM include cholesteatoma, mastoiditis, facial nerve paralysis, lateral sinus thrombophlebitis, meningitis, and brain abscesses [7]. CSOM has been suggested to be responsible for up to 80% of acquired hearing impairment in some communities, with reports showing that between 40 and 91% of children have some degree of permanent hearing loss [6,8]. This can translate into difficulty with speech acquisition, impaired cognitive development, poor school performance, and difficulty in finding future employment in adults, and carries an associated social stigma.

South Africa sits at a unique cross section of the developing and developed world, providing a unique opportunity to refine standard of care approaches to diseases of poverty. CSOM is treated routinely by primary care physicians, but complicated or refractory disease is managed by otolaryngologists (ear, nose and throat surgeons, or ENTs). Despite struggles with budgets and infrastructure, the health care system in South Africa engages in state-of-the-art ENT surgical practice, including resection of head and neck cancer with the use of free flap reconstruction, cochlear implantation, and endoscopic skull base surgery [9,10]. Only 0.9% of registered medical practitioners in South Africa are ENT physicians, and in many cases a single ENT cares for a population of several million; a subset of the population (~31 million South Africans) are served by less than 35 full time ENT specialists [9]. South Africa faces additional challenges, having the largest HIV positive population of any country, at an estimated 7.1 million by the Joint United Nations Programme on HIV/AIDS (UNAIDS) reports.

There have been a few studies in urban centers in South Africa aimed at investigating chronic otorrhea. Meyer et al. reported on 79 patients presenting with chronic otorrhea to an outpatient ENT clinic at an urban hospital in Cape Town, South Africa [5]. They found a high prevalence of *Proteus mirabilis*, and most of the isolated organisms were sensitive to fluoroquinolones. Tiedt et al., conducted a similar study of CSOM in 86 children at the ENT clinic at Universitas Academic Hospital in the Free State Province [11]. They also identified a preponderance of Gram-negative bacteria, of which 95% were sensitive to fluoroquinolones. In this study, 50% of the children were HIV positive and two-thirds demonstrated hearing loss to some degree.

To extend these studies and evaluate the role of routine culture in the care of at-risk patients with CSOM, we evaluated a rural population in the KwaZulu-Natal Province of South Africa. The Msinga sub-district in which we worked has a population of ~160,000 people, predominantly Zulu (99%), and suffers from a markedly high prevalence of HIV (~30% of antenatal patients) and tuberculosis (TB) incidence rate (>1100/100,000). Given that a majority of the CSOM literature has been sourced from urban centers, which typically have better access to subspecialist care and onsite or nearby microbiology infrastructure, this population offers insight into the utility of using remote microbiology services as a means of augmenting care for patients in rural areas. Additionally, this study serves as a pilot for evaluating pathogens and resistance patterns in this unique landscape dominated by high endemic rates of TB and HIV. We report both expected and unexpected pathogens, and are able to use culture results to directly impact patient care for patients who had been suffering from CSOM for years and had received multiple courses of empiric therapy.

## 2. Methods

We established a temporary clinic in Tugela Ferry, South Africa to conduct a needs assessment of ENT pathology in rural KwaZulu-Natal, South Africa. Patients were recruited by means of advertisement and word of mouth in satellite clinics across the Msinga municipality. Msinga is an ~2500 km^2^ municipality with a population of ~160,000 [12]. The advertisements recruited patients with pain, infection, swelling, or other problems involving the head or neck, and gave a list of common ENT conditions and symptoms, including symptoms of ear pain, drainage from the ear, or decreased hearing. This was posted in local clinical spaces in both English and a certified Zulu translation. Over 400 patients were seen in this clinic over five weeks. Patients were triaged for the presence of otorrhea on otoscopic exam, and were eligible to be included in the study if they had had ear drainage for >2 weeks.

The purpose and details of the study were explained by a research assistant fluent in Zulu and English. Informed consent and assent (as appropriate) were obtained from patients or their parents or guardians in either English or a certified Zulu translation. Additional information regarding age, home address, duration of otorrhea, prior antibiotic regimens, and HIV status were collected. If consent was obtained, the external ear and entry to the external auditory canal (EAC) was cleaned with an alcohol swab, a sterile ear speculum was placed, and a sample from an area of purulence was collected using a narrow swab by an ENT provider. As needed, suctioning of the ear was performed to remove debris and EAC contaminants, in order to reveal underlying infection prior to these steps. In cases of bilateral drainage, cultures were obtained from each ear. These samples were sent to the South African National Health Laboratory Systems for bacterial, fungal, and mycobacterial culture, and analyzed using standard laboratory techniques, with routine antimicrobial susceptibility testing performed and interpreted using guidelines from the Clinical and Laboratory Standards Institute (CLSI).

In all cases where changes to initial therapy were indicated by culture, but the patient was not available for follow-up, efforts were made to contact the patient and recommend follow-up with local practitioners.

Standard descriptive statistics were used, including a *t*-test, ANOVA, and Chi-square analysis using GraphPad Prism (version 7.0a, www.graphpad.com), which was used for the preparation of figures. A *p*-value of <0.05 was considered significant for all data endpoints.

### Ethical Review

This study was reviewed and approved by both the Yale Institutional Review Boards for Human Research (HIC#1510016604, 5 November 2015) and the South African Medical Association Research Ethics Committee (ENTID2015, 12 February 2016). 

## 3. Results

Over five weeks, approximately 400 patients from the Msinga municipality of KwaZulu-Natal, South Africa were screened as a part of a needs assessment for ENT services and CSOM in rural South Africa. From this cohort, a total of 14 patients were enrolled. Three additional patients met study criteria but did not consent to the study. This suggests a prevalence estimate from this population of ~4%. Eighteen swabs were collected (four patients had bilateral disease) from areas of purulence via a sterile speculum to minimize contamination with colonizing bacteria. (Figure 1) Five of the patients were female (36%) and nine were male (64%). The average age of the patients was 22 ± 19 years; eight of the patients were under 18 years of age. Six patients (43%) were known to be HIV-positive, while the others were untested.

A variety of pathogens were cultured, but Gram-negative organisms predominated. *Staphylococcus aureus* (*n* = 5), however, was the most commonly isolated organism, followed by *Pseudomonas aeruginosa* (*n* = 3), *Proteus mirabilis* (*n* = 2), and *Providencia stuartii* (*n* = 1). Five patients had no pathogenic bacteria or no growth on culture. One patient grew a *Scedosporium* species. (Figure 2).

A majority of the recovered organisms were sensitive to most of the commonly prescribed antibiotics tested. Among the Gram-positive organisms, only the *Staphylococcus aureus* isolates had antibiotic sensitivities reported. Three of these isolates exhibited broad sensitivity, but two cases of methicillin-resistant *Staphylococcus aureus* (MRSA) were cultured (bilateral specimens from a single patient). Very little resistance was seen for the Gram-negative isolates (Figure 3).

Of the six patients with more complete medical records, due to longitudinal care for HIV, three were known to have significantly reduced CD4 counts and no patients were known to have achieved virological suppression. The amount of drainage varied between patients, but five of these patients had perforation or complete destruction of the tympanic membrane. Only two patients had not had prior treatment for CSOM.

All patients seen in consultation associated with this study were prescribed amoxicillin/clavulanic acid and topical treatment as available. Amoxicillin/clavulanic acid is a routinely available antibiotic, which satisfied the empiric coverage recommendations for therapy for CSOM in South Africa. Topical therapy was dictated by the availability of ofloxacin. In cases of shortage, acetic acid was used. Dexamethasone drops were used in all cases.

Multiple patients either walked or were bussed in from distant locations, and were typically not available for close follow-up. The six patients with robust medical records, who were existing patients seen in the HIV/TB clinic or admitted as inpatients at the nearby Church of Scotland Hospital, were seen in follow-up 1–2 weeks after their initial visit. For these patients, adjustments to therapy were made based on culture results. (Table 1) In four out of six of these cases, changes to empiric therapy were made based on the culture results. All patients were referred to audiology and counseled on dry ear precautions. In the three cases where follow-up occurred after appropriate therapy (either empiric or after changes made based on culture), drainage was notably decreased.

Regarding examination for tuberculosis as an etiology of CSOM in the region, twelve mycobacterial cultures returned without any TB identified. However, one patient whose culture returned without growth had classical features of tuberculous CSOM—painless otorrhea, thick exudate, multiple small tympanic membrane perforations, and a close contact who had TB—and was referred for follow-up.

## 4. Discussion

CSOM is a serious and yet treatable disease of poverty that can affect up to 20% of children and adults in at-risk populations, and is the cause of up to 80% of acquired hearing loss globally [6]. This survey identified CSOM in approximately 4% of the screened population, consistent with literature reports and the idea that this condition places a significant and under-appreciated burden on at-risk communities. Though classically associated with *Pseudomonas aeruginosa, Staphylococcus aureus* and *Klebsiella* spp. infection, the responsible pathogens may vary depending on local factors, including comorbidities and patterns of antibiotics use [4]. In South Africa, treatment is typically empiric, and cultures are not routinely obtained to guide therapy. Studies of the microbiology of CSOM in Southern Africa or in populations with a high prevalence of HIV have been limited.

Notably, South Africa has a robust healthcare infrastructure, including ENT specialists, but access to these specialists is largely concentrated in urban areas. In more rural areas, it is not uncommon for ENT providers to care for a catchment area of millions of people—a relatively common feature among Low and Middle Income Countries countries [10]. In cases of severe complications of CSOM, the referral and transfer of patients from remote rural areas is possible. However, efforts to make a significant impact on the treatment of CSOM in general will likely need to be focused at the primary care level.

We sought to evaluate the utility of recommending routine culture be obtained from all patients with CSOM, as a means of improving treatment of this disease at the time of initial point of contact with the healthcare system.

In this proof of concept study, we were able to use cultures obtained from our rural population to directly impact patient care. We made changes to the initially prescribed empiric antibiotic therapies in four out of six of the patients for whom we were able to schedule follow-up within the timeframe of the study. All patients had ongoing follow-up with local providers scheduled independent of the study. In patients whose therapy was appropriate for their culture results and who were able to be seen in follow-up, we noted significant improvement in drainage, even from patients who had symptoms lasting for years and had received multiple lines of prior antibiotics.

Routinely obtaining a culture comes with an additional added cost, however. Tiedt et al. reported that the average duration of CSOM in their study in a South African ENT clinic was 162 weeks [11]. This is consistent with our experience that patients often have multiple visits and treatment courses for CSOM over an extended period of time. In this setting, the cost of an initial culture may be more than compensated for by decreasing the need for multiple visits and treatment courses. According to the 2013 South African National Health Laboratory Service price list, the cost of an aerobic and anaerobic culture was R49.34 and R35.37, respectively [13]. This compares with estimates of the average cost of a course of a beta-lactam antibiotic at R99.63 in a 2010 review of a national community pharmacy group in South Africa that was felt to be representative of a typical community pharmacy [14]. While these numbers are not completely indicative of the cost to the government health care system that cares for the vast majority of the at-risk communities, which bear the brunt of CSOM, they give a sense of the relative costs involved, and support the idea that routine culture for CSOM is cost-effective. The proof of concept study presented here demonstrates that routine culture can inform treatment, especially in cases of resistant or atypical pathogens, which may make up a substantial portion of the organisms driving this disease process. Taken together, these questions could form that basis for additional larger scale and more comprehensive studies.

It is also notable that even in a community where we would expect relatively little antibiotic selection pressure, we identified one patient with *MRSA*, one with *Scedosporium* species, and one with clinical features of tuberculosis CSOM (painless otorrhea, thick exudate, multiple small tympanic membrane perforations, and close contact who was TB positive). In these patients, even despite the occasionally limited range of therapies available at our clinic, we were able to find appropriate interim treatment for these more resistant pathogens and arrange for follow-up with ENT. Triaging only the more complicated or resistant cases to ENT specialists, while providing appropriate care for uncomplicated CSOM at the primary care level, is likely to prove a successful strategy for providing better care for this disease overall, preventing complications, and decompressing ENT clinics to allow for management of complicated cases.

We found one patient with cultures growing a *Scedosporium* species. *Scedosporium* is associated with limited access to clean water, can be cultured worldwide from sewage and soil, and can cause a variety of infections in immunocompetent and immunocompromised patients [15,16]. At the case report level, it has been noted to cause otomycosis [15,16,17,18]. The patient in this study was HIV positive, although had no recent labs available for review, and had had a protracted course of CSOM treated with eight courses of various oral and topical therapies over the course of four years; this prolonged course is in keeping with the limited reports in the literature [15,16,17,18].

Treatment of *Scedosporium* is difficult given its relative resistance to antifungals; however, it is often susceptible to voriconazole. There are two case reports of successful treatment with topical clotrimazole and surgical debridement [18]. Our isolate did not have sensitivities tested. Voriconazole was not available in our clinic, but we were able to offer topical clotrimazole in combination with suctioning, dry ear precautions, ENT referral, and counseling regarding adherence to an antiviral regimen as initial treatment [15]. When seen in follow-up, that patient noted a significant reduction in drainage.

In the patient with MRSA CSOM, topical mupirocin was applied in additional to dry ear precautions and a referral to ENT [19]. On follow-up, drainage was decreased.

This study has several limitations. Not all patients had robust medical records, allowing for detailed analysis in only six of our patients. The small sample size and lack of routine follow-up for patients precludes drawing statistically significant conclusions; however, this does serve as a proof of concept study that demonstrates that culture-based care is possible in rural locations where the infrastructure to process cultures exists, even if the microbiology labs are remote. Notably, delays in processing cultures may result in loss of some culture information; however, this study demonstrates that useful data can be obtained even with delays of up to 48 h from sample collection to culture, as was the case in our study.

Our sample population had a recruitment bias towards patients who were able to travel to our clinic, and disproportionately included patients who had already established care in the existing HIV and tuberculosis clinics. This misses patients with more difficulty accessing healthcare.

We made significant efforts to minimize contamination from skin colonizers through sterile techniques—however, one must consider the possibility that the recovered organisms were colonizing the area, and were not contributing to infection. In each of the eight cultures where the amount of growth was quantified, it was reported as “heavy”, more consistent with infection than colonization. A standardized approach to the collection of these samples that includes a conscientious attempt to limit the recovery of colonizing bacteria and improve the yield of the organisms contributing to the infection is critical to this kind of research. We suggest that suctioning of superficial debris be performed as needed to reveal the underlying infection, after which the external ear and inlet to the EAC be cleaned with an alcohol swab prior to collection of the culture. Samples should be obtained from area of purulence, and we recommend the culture swab be introduced through a sterile speculum.

This was a small pilot study, but the first to be conducted in a primary care clinic in Southern Africa. We encountered both expected and atypical pathogens, including a rarely reported *Scedosporium* isolate, and demonstrated a role for routine culture at the initial point of healthcare contact for patients with CSOM.

Moving forward, we are seeking to expand this model to launch a province-wide survey of the microbiology of CSOM and further investigate the role of HIV and TB in this disease process. We hope that this will continue to raise awareness of this devastating infection and help inform the standard of care approach to treatment.

## Figures and Tables

**Figure 1 tropicalmed-04-00010-f001:**
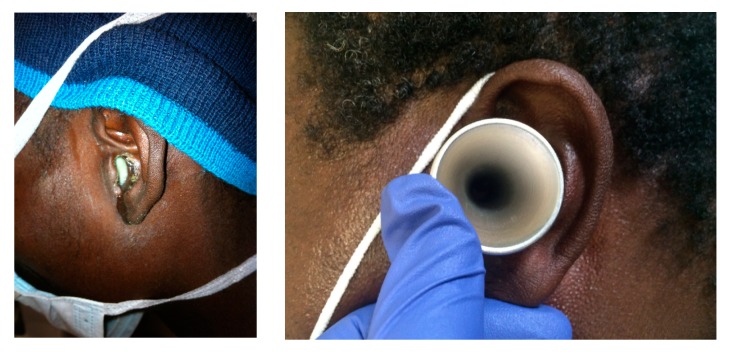
Chronic Suppurative Otitis Media (CSOM) presentation and collection of samples. Example of purulent drainage visualized in the external auditory canal (EAC) of a patient with CSOM. Samples were obtained by suctioning the most superficial material as needed, cleaning the external ear and entrance of the EAC with an alcohol swab, and then obtaining the culture from an area of visualized purulence through a sterile speculum.

**Figure 2 tropicalmed-04-00010-f002:**
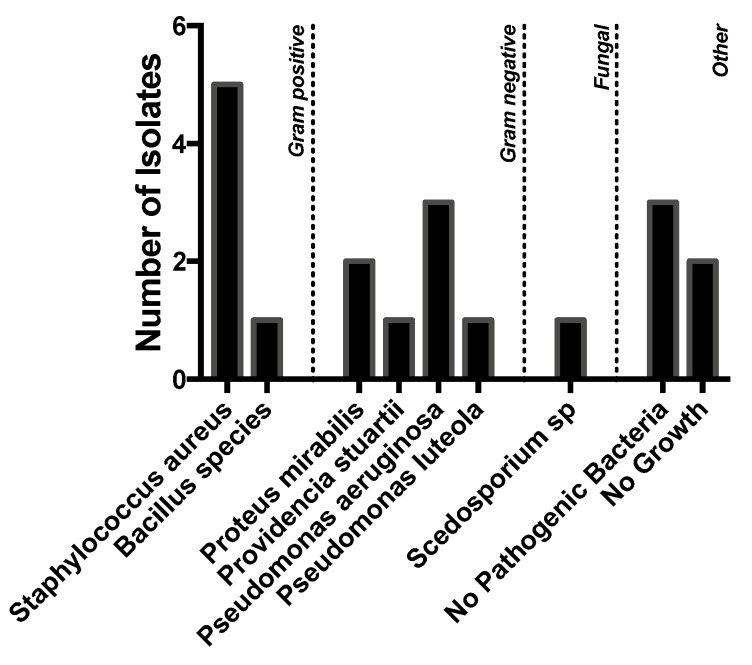
Number of positive swabs. Number of isolates of organisms obtained by culture of purulent drainage in patients with CSOM.

**Figure 3 tropicalmed-04-00010-f003:**
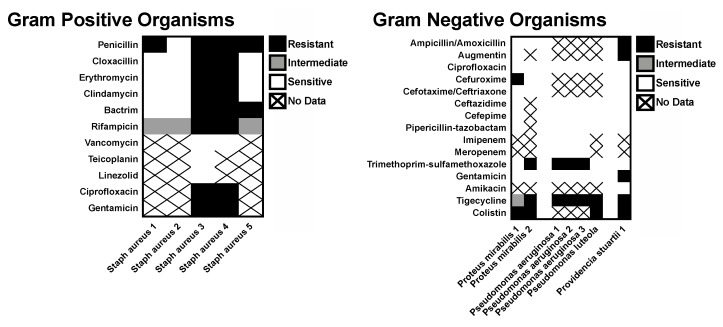
Antibiotic resistance. Map of resistance of isolates obtained by culture of purulent drainage in patients with CSOM.

**Table 1 tropicalmed-04-00010-t001:** Patient presentation and management. Presenting characteristics, exam findings, prior therapy, initial study therapy, culture results, and alterations in therapy based on culture results for the six patients with prior medical records.

Patient ID (Age/Sex)	HIV Status	Presenting Symptoms	Exam	Prior Treatment	Initial Treatment	Culture	Follow-up
1 (15-year-old M)	Positive VL: 1223 CD4: 791	Ear pain Hearing loss	Perforated tympanic membrane. Light ear drainage	Multiple oral and topical treatments	Amoxicillin/clavulanic acid	*P. aeruginosa*	Ongoing drainage Changed to ciprofloxacin drops
2 (41-year-old F)	Positive VL: 980761 CD4: 13	Right ear drainage Admitted for failure to thrive	Heavy ear drainage	None	Initially on ceftriaxone Added ofloxacin and dexamethasone drops	*MSSA*	Decreased drainage Continued therapy
3 (44-year-old F)	Positive VL: Unknown CD4: 10	Right ear pain Bell’s palsy Admitted for failure to thrive	Perforated tympanic membrane Light ear drainage	None	Amoxicillin/clavulanic acid Acetic acid and dexamethasone drops	No pathogens isolated	No major change Continued therapy
4 (3-year-old F)	Positive VL: 2613 CD4: 78	Right ear drainage	No residual tympanic membrane	Acetic acid drops	Amoxicillin/clavulanic acid Acetic acid and dexamethasone drops	*P. aeruginosa*	Ongoing drainage Changed to ciprofloxacin drops
5 (55-year-old M)	Positive No recent available testing	Long standing drainage Hearing loss	No tympanic membrane Light ear drainage	Framycetin sulfate, gramicidin, and dexamethasone drops	Amoxicillin/clavulanic acid PO dexamethasone drops and acetic acid drops	*Scedosporium* species	Topical Clotrimazole followed by Ofloxacin and dexamthasone drops 2nd Follow-up–Decreased drainage
6 (33-year-old F)	Positive VL: 733 CD4: 573	Long standing drainage Hearing loss	40% perforation	Ofloxacin and dexamethasone drops	Amoxicillin/clavulanic acid Ofloxacin and dexamethasone drops	*MRSA*	Ongoing drainage Mupirocin ointment ofloxacin and dexamethasone drops 2nd Follow-up–Decreased drainage

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
