# Peer review of "The Role of Routine Culture in the Treatment of Chronic Suppurative Otitis Media: Implications for the Standard of Care in Rural Areas of South Africa"

_tropicalmed, 2019, doi:10.3390/tropicalmed4010010_

Round 1
Reviewer 1 Report
May address the EAC contamination problem when obtaining culture
Put some thoughts on the cost-effectiveness issue while routine doing the culture
May refer to national or regional bacterial/resistance data in order to support the need for routine ear culture
Author Response
1. May address the EAC contamination problem when obtaining culture
We agree that EAC contamination is a significant consideration for this kind of research. To minimize this, all cultures were obtained after cleaning the EAC with an alcohol swab. Cultures were obtained through a sterile speculum from areas of purulence. As needed, suctioning of the ear was performed to remove external debris and EAC contaminant to reveal underlying infection. Additionally, we evaluated the amount of growth reported for each culture when this information was available; this was reported in 8 of the 14 cultures where an organism was cultures. For each of these cultures, the growth was reported as “Heavy”, more suggestive of pathogenic organism than colonization. Ideally, the gram stain could also be evaluated for the number of WBCs in the culture to further support the idea that a culture was obtained from areas of inflammation, however, this information was not routinely reported for this set of cultures by the South African National Health Laboratory System. As we endeavored to evaluate the use of the microbiological infrastructure as it currently exists, rather than under artificial research conditions where each culture was processed and evaluated in detail, we accept this as an obvious limitation of our study. These ideas have been expanded on in detail in the methods and discussion.
2. Put some thoughts on the cost-effectiveness issue while routine doing the culture
We agree that this strengthens the paper and have included the following in the discussion to address this point.
“Routinely obtaining a culture comes with an additional added cost, however, Tiedt et al reported that the average duration of CSOM in their study in a South African ENT clinic was 162 weeks. This is consistent with our experience that patients often have multiple visits and treatment courses for CSOM over an extended period of time. In this setting, the cost of an initial culture may be more than compensated for by decreasing the need for multiple visits and treatment courses.
According to the 2013 South African National Health Laboratory Service price list the cost of an aerobic and anaerobic culture was R49.34 and R35.37 respectively. This compares with estimates of the average cost of a course of a beta-lactam antibiotic at R99.63 in a 2010 review of a national community pharmacy group in South Africa that was felt to be representative of a typical community pharmacy. While these numbers are not completely indicative of the cost to the government health care system that cares for the vast majority of the at-risk communities which bear the brunt of CSOM, they give a sense of the relative costs involved.
The proof of concept study presented here demonstrates that routine culture can inform treatment, especially in cases of resistant or atypical pathogens, which may make up a substantial portion of the organisms driving this disease process. Taken together, these questions could form that basis for additional larger scale studies.”
We have added the following references to support this new paragraph:
Truter, I. (2015). Antimicrobial prescribing in South Africa using a large pharmacy database: A drug utilisation study Ilse Truter. Southern African Journal of Infectious Diseases. http://doi.org/10.1080/23120053.2015.1054181
National Health Laboratory System State Price List. Availabe online: http://www.health.gov.za/index.php/shortcodes/2015-03-29-10-42-47/2015-04-30-09-10-23/uniform-patient-fee-schedule/category/221-u2014?download=895:nhls-state-price-list-2013-annexure-m (accessed on 12/20/18).
3. May refer to national or regional bacterial/resistance data in order to support the need for routine ear culture
Infectious diseases remains a major health care consideration in Southern Africa which is compounded by the twin epidemics of HIV and tuberculosis. Detailed data on antimicrobial resistance is available for certain pathogens and infectious syndromes but a comprehensive picture of antimicrobial resistance has yet to be established in Southern Africa. This study, while small demonstrates the utility of using routine culture for the treatment of CSOM but also suggests that larger scale screening efforts for CSOM could add an important dimension to the existing understanding of the state of antimicrobial resistance in Southern Africa.
There is a paucity of literature describing resistance rates in CSOM. The cited studies by Meyer and Tiedt in the manuscript describe the existing literature in South Africa. There is a body of literature describing increasing pneumococcal antimicrobial resistance with rates of penicillin resistance of >75%. Resistance to erythromycin and clindamycin in S. pneumonia has been reported at 25% and >90% respectively. Additionally, rates of resistance to co-trimoxazole have been reported as high as 72%. Haemophilus influenzae is another respiratory pathogen which often contributes CSOM and beta-lactam resistance has been reported at greater to 45%. These rates of resistance support the role of culture in the treatment of CSOM.
Crowther-Gibson, P., Govender, N., Lewis, D. A., Bamford, C., Brink, A., Gottberg, von, A., et al. (n.d.). Part IV. Human infections and antibiotic resistance. SAMJ: South African Medical Journal, 101(8), 567–578.
Reviewer 2 Report
An important study to determine the value and feasibility of routine microbiology for management of CSOM in regional South Africa. Interesting study that reads well.
1. L64 “refractory” rather than “refractive”
2. Capitalise “Gram”.
3. Please provide greater detail and/or references on culture methods and identification of microbes.
4. L100. Please provide more detail on, “Prevalence of 4%”. What is the population? E.g. was the advert (L235-236) for people with any form of ear disease?
5. L100. 14 patients plus 5 bilaterals doesn’t equal 18 swabs.
6. L111. Please don’t use term “relative” sensitivity. Just report as per CSLI breakpoints.
7. Figure 2. Suggest title to be Number of swabs positive.
8. L106-113. Suggest a little more discussion of the data to guide readers through figures 2 and 3.
9. Table 1 is missing.
10. L186-189. References needed.
11. L215. “existing”
12. L219. Suggest changing “pathogenic” to “contributing to the infection”.
13. Discussion. For implementation into routine practice, are there technical recommendations that you can make based on your findings?
Author Response
1. L64 “refractory” rather than “refractive”
Changed as recommended
2. Capitalise “Gram”.
Changed as recommended
3. Please provide greater detail and/or references on culture methods and identification of microbes.
We have expanded on the method of collecting the samples to improve yield of organism contributing to infection rather than colonizing bacteria from the External Auditory Canal (EAC) as EAC contamination is a significant consideration for this kind of research. To minimize this, all cultures were obtained after cleaning the EAC with an alcohol swab. Cultures were obtained through a sterile speculum from areas of purulence. As needed, suctioning of the ear was performed to remove external debris and EAC contaminant to reveal underlying infection.
Additionally, we have indicated that the samples were submitted to the South African National Health Laboratory System for microbiological analysis. The NHLS is the sole provider to the public sector of South Africa for diagnostic pathology and covers ~80% of the population.
4. L100. Please provide more detail on, “Prevalence of 4%”. What is the population? E.g. was the advert (L235-236) for people with any form of ear disease?
Thank you for identifying this lack of clarity. We have expanded on the nature of the needs assessment work from which this cohort was derived.
5. L100. 14 patients plus 5 bilaterals doesn’t equal 18 swabs.
Typo corrected.
6. L111. Please don’t use term “relative” sensitivity. Just report as per CSLI breakpoints.
Thank you for identifying this imprecision in language. All items were reported as either sensitive or resistant per CLSI guidelines. This statement was meant to reflect that overall the organisms were sensitive to multiple antibiotics. We have amended the language to reflect this.
7. Figure 2. Suggest title to be Number of Swabs Positive.
We agree with this suggestion and have amended the title.
8. L106-113. Suggest a little more discussion of the data to guide readers through figures 2 and 3.
Figure legends appear to have not loaded on the initial submission. These have been added to the update manuscript and we have described these results in more detail in the manuscript text as well.
Figure Legend I:
Example of purulent drainage visualized in the external auditory canal (EAC) of a patient with CSOM. Samples were obtained by suctioning the most superficial material as needed, cleaning the external ear and entrance of the EAC with an alcohol swab, and then obtaining the culture from an area of visualized purulence through a sterile speculum.
Figure Legend II:
Number of isolates of organisms obtained by culture of purulent drainage in patients with CSOM.
Figure Legend III:
Map of resistance of isolates obtained by culture of purulent drainage in patients with CSOM.
9. Table 1 is missing.
Table 1 appears to have not loaded on the initial submission. This has been added to the updated manuscript with our apologies.
10. L186-189. References needed.
The references have been updated.
11. L215. “existing”
Changed as recommended.
12. L219. Suggest changing “pathogenic” to “contributing to the infection.”
Changed as recommended.
13. Discussion. For implementation into routine practice, are there technical recommendation that you can make based on your findings?
We are underpowered in this study to make conclusive statements about treatment and outcomes, however, this point is well made. We think that a standardized approach to the collection of these samples that includes a conscientious attempt to limit recovery of colonizing bacteria and improve yield of the organisms contributing to the infection is critical to this kind of research. We suggest that the external ear and inlet to the EAC be cleaned with an alcohol swab and suctioning of superficial debris and material as needed to reveal underlying infection. Samples should be obtained from area of purulence and we recommend the culture swab be introduced through a sterile speculum.
We have added this to the discussion.
Round 2
Reviewer 1 Report
Suggest accept
Author Response
Dear Sir or Madam,
Thank you for you time and consideration.
Warmest regards,
J. Zachary Porterfield